# The Potential Value of Debarking Water as a Source of Polyphenolic Compounds for the Specialty Chemicals Sector

**DOI:** 10.3390/molecules28020542

**Published:** 2023-01-05

**Authors:** Kelly Peeters, Esakkiammal Sudha Esakkimuthu, Črtomir Tavzes, Katja Kramberger, Ana Miklavčič Višnjevec

**Affiliations:** 1InnoRenew CoE, 6310 Izola, Slovenia; 2Andrej Marušič Institute, University of Primorska, 6000 Koper, Slovenia; 3Faculty of Health Sciences, University of Primorska, 6310 Izola, Slovenia; 4Faculty of Mathematics, Natural Sciences and Information Technologies, University of Primorska, 6000 Koper, Slovenia

**Keywords:** bark, debarking water, antioxidant activity, LC-MS-MS polyphenolic compounds

## Abstract

Forest-based industries produce huge quantities of bark during their primary processing activities. In Nordic pulp and paper industries, where a wet debarking process is used for bark removal, toxic debarking water and bark press water are produced as a by-product. However, polyphenols represent an important fraction of the debarking water and bark press water. These polyphenolic compounds are of commercial interest in chemical specialty sectors since polyphenols have been proven to have diverse health benefits, and after collecting them from waste sources, they can act as alternatives to oil-based chemicals. Determining the economic potential of polyphenolic compounds, identifying their molecular structure, and determining the antioxidant capacity of these compounds present in debarking water and bark can support the identification of their potential applications. The results show that water extractions from bark have a lower efficiency than (partial) alcoholic extractions. Nevertheless, a considerable amount of low-molecular polyphenolic compounds, which are of interest for high-end applications, was found in all extracts. Bark press water has a highly versatile range of polyphenolic compounds and showed some antioxidant activity, making it a great source for the collection of polyphenolic compounds, in contrast to debarking water, which had a much lower polyphenolic content and low antioxidant activity.

## 1. Introduction

In 2020, the European pulp and paper industry used close to 146 million cubic meters of wood [1]. These forest-based industries consequently produce huge quantities of bark during their primary processing activities. Namely, since bark reduces the quality of wood products, it needs to be removed. In Nordic pulp and paper industries, a wet debarking process is used for bark removal, producing toxic debarking water as a by-product. For each cubic meter of wood, up to 2 m^3^ of water is used for debarking [2]. Due to the toxicity, the mill must treat the debarking water in a costly process. However, this otherwise toxic water also contains useful bioactive compounds. About 20–40% of the dry weight of the bark consists of lipophilic and hydrophilic extractives [3], which could potentially be used in a wide range of high-value applications. Of these compounds, polyphenols are present in large amounts. Kylliäinen and Holmbom [4] investigated debarking water and bark press water compositions from the Metsä-Serla mill in Lielahti and found that polyphenols represented a significant fraction of the debarking water. They found that stilbene glucosides accounted for 14%, and polyphenols for 12%, of the total organic content in the debarking water. These polyphenolic compounds are of commercial interest in the pharmaceutical, cosmetic, and functional food sectors since they can act as alternatives to oil-based chemicals. This interest arises from the fact that polyphenols have been proven to have diverse health benefits, such as anti-inflammation, antioxidation, signal transduction modulation, anti-scavenging, anti-proliferation, and anti-microbial activities [5]. They also play important roles in organoleptic properties, such as the sharpness, bitter taste, colour, and turbidity of various food products such as wine and cider [6]. Finding methods to remove large amounts of these extractives from debarking water and bark press water is of great environmental importance and presents an economic opportunity when reused for other applications [7]. However, the identification of the molecular structure of the polyphenolic compounds present in these two sources is first needed to identify their potential applications. Furthermore, the determination of the antioxidant capacity can play an indicative role in the usefulness of these wastewaters as a source of polyphenols. Antioxidant assays play a crucial role in the high-throughput and cost-effective assessment of the antioxidant capacities of natural products [8]. Numerous methods have been developed, beginning with α, α-diphenyl-β-picrylhydrazyl (DPPH) free radical scavenging [9], but several investigators have expressed concerns about their reliability [8,10,11]. Such concerns arise mainly from the poor correlation between in vitro and in vivo results [8]. However, the hydrogen atom or electron-donating capacity data obtained by these methods have provided important information on their intrinsic antioxidant potential, with minimum environmental interference [11]. This offers the ability to correlate the polyphenolic quantities measured by LC-MS-MS with DPPH results and enables a more calculated decision about which samples to include for further in vivo studies. An investigation into the interplay of the antioxidant capacity, the total phenol content, tannins, and the cytotoxic and antiproliferative activity of aqueous extracts showed that cytotoxic effects on cancer cells were closely related to the antioxidant capacity [12].

Many research groups have tackled various issues connected to polyphenolic compound extraction, characterisation, and antioxidant activity and its implications. New techniques were developed for the structural elucidation of stilbenes with HPLC and NMR [13]. Different extraction methods were compared (pressurised fluid extraction performed the best) for their influence on the obtained antioxidant activity and the polyphenolic composition (NMR and HPLC-DAD-MS/MS, HPLC) [14,15]. Supercritical carbon dioxide extraction was also used for the extraction of polyphenolic compounds from bark, and the optimisation of the parameters was related to the obtained total polyphenolic content and antioxidant activity [16]. The parameters for accelerated solvent extraction were optimised to determine the polyphenolic content (characterisation with FTIR and GC-MS) [17]. Ghitescu et al. [18] optimised the ultrasound-assisted extraction procedure of spruce bark with water and ethanol based on the total phenol content. Bark extracts (Soxhlet extraction with polar and apolar solvents) were used to improve the oxidation stability of lard. This characteristic was correlated to the polyphenolic composition (GC-MS) and antioxidant capacity [19]. Since polyphenolic compounds play an important role in the resistance against various biotic and abiotic stresses, monomeric and dimeric flavan-3-ols were analysed by HPLC-MS/MS, including its degree of polymerisation in relation to fungal pathogenic attack studies in Norway spruce [20]. The parameters for hot water extraction of spruce bark were optimised, and the polyphenolic compounds were qualified and quantified with HPLC [21], or HPLC, MS, and NMR [22]. This was carried out to understand the anti-inflammatory and nutraceutical potential of the extract. Inner and outer bark extractions with solvents of different polarities were performed to determine the stilbene glucoside and tannin contents with GC-MS analysis and HP-SEC, yielding knowledge of their possible utilisation in chemical processes [23]. Since adhesive properties are dependent on the tannin structure, the structures of condensed tannins in aqueous extracts of spruce bark tissue were analysed using MALDI-TOF mass spectrometry, HPLC-DAD, HPLC-MS, and the total phenolic content was determined by Folin–Ciocalteu assay [24,25,26]. Extensive reviews about spruce bark polyphenolic composition have been prepared by Routa et al. [27] and Tanase et al. [28]. Research on the polyphenolic composition of debarking water and bark press water is scarcer. Multia [29] investigated stilbene glucosides in the aqueous extracts of spruce bark, debarking water, and bark press water. Kylliäinen and Holmbom [4] analysed the major dissolved and colloidal components in the water extracts of Norway spruce bark and debarking effluents from pulp mills. They found that small amounts of free stilbene, catechin, and various tannins constituted about one-third of the organic material present therein.

The aim of this research was to directly compare different bark extraction techniques (hot water, cold water, and ethanol) to ascertain the differences in the types of polyphenolic compounds and their quantity obtained in water and ethanolic extracts. The focus was on the estimation of water’s efficiency as an extraction solvent for simple polyphenolic compounds, as this can have a profound effect on the usability of the extracts for other industries. Measuring the polyphenolic concentrations in the debarking water and bark press water can indicate if these by-products can be a potential polyphenolic source for the speciality chemical industry, and not only for low-grade applications (e.g., large tannin and lignin compounds for glue or paints), i.e., if they contain high-enough concentrations of small polyphenolic compounds that did not undergo polyphenolic transformations (e.g., oxidation). Therefore, antioxidant capacity was correlated to the polyphenolic concentration to ascertain the level of oxidation that occurred in the samples and to evaluate which extracts have potential for further research on their effects on isolated cells in vivo. While studies on polyphenols in debarking water are scarce, it is of the greatest importance to study and evaluate the necessary quality of polyphenols found in by-products for their further use in the food, cosmetics, and pharmaceutical industries [30].

## 2. Results and Discussion

In all the tested sample sources (debarking water, bark press water, and bark extracts), 25 phenolic compounds were determined by HPLC-DAD-MS/MS. A total of 17 compounds were identified by a targeted MS analysis, of which only 8 of the compounds were in a high enough quantity to be detected by the DAD: quinic acid, protocatechinic acid, catechin, ferulic acid, dihydrorobinetin, quercitrin, spiraeoside, isorhamnetin rhamnoside. Another eight other DAD peaks, which could be linked to polyphenolic compounds, remain unknown. A possible matching identity was sought in accordance with the literature according to *m/z*. However, the fragmentation patterns were not correlated, so it is possible that different isomers of these compounds were detected. Although similar compounds were determined in different samples, the quantity of these compounds differed according to the sample source used.

### 2.1. Identification of the Polyphenolic Compounds by HPLC-DAD-MS/MS

Debarking water, bark press water, and bark extracts were measured by HPLC-ESI-Q-TOF-MS and the elution of polyphenolic compounds was followed with DAD analysis according to their retention time (see Figure 1). With mass spectrometry, the identities of the main peaks observed by DAD were determined. For all samples, peaks at the same retention times were observed, but they differed in intensity.

Targeted analysis confirmed the identity of eight peaks (see Table 1), which were also detected by the DAD. These peaks are indicated in Figure 1 with black numbers (1, 2, 3, etc.). Quinic acid, protocatechuic acid, catechin, ferulic acid, dihydrorobinetin, quercitrin, spiraeoside, and isorhamnetin rhamnoside were found. Their fragmentation patterns were confirmed with the fragmentation patterns found in the literature [31,32,33,34,35].

An analysis of the unknown discernible peaks determined by the DAD detector, other than those shown in Table 1, was also performed. These peaks are indicated in Figure 1 with blue numbers (1a, 2a, 3a, etc.). The peaks were further characterised by MS/MS according to the retention time, exact mass (*m/z* ratio), and fragmentation pattern (Table 2), similar to the targeted analysis. However, the following attributions were made based on only the *m/z* ratios found in the literature in comparison to the measurements, but their fragmentation patterns differed (or were not found in the literature sources), so the proposed compounds are rather a guideline and a possible isomer of the mentioned compounds. Peak 1a was identified with *m/z* 371.097 as oxomatairesinol. The compound was detected before by Mansikkala et al. [36] in knotwood of Norway spruce. Peak 2a could not be identified in the literature. Peak 3a was identified as (epi)catechin-o-glucoside, a compound that was found in the bark of Scots Pine [37]. Compound **4a** with *m/z* 441.0954 was identified as (epi)catechin monogallate, a compound that was found in pine by-products [38]. Compound **5a** with *m/z* 525.1978 was expected to be isorhamnetin glucoside, a compound that was found in bark from Abies alba [39]. Compound **6a** with *m/z* 373.1506 was identified as (iso)hydroxymataireresinol, a compound that was found before in Norway spruce knotwood [37] and bark [40]. Peak 7a with *m/z* 647.1764 is expected to be a piceatannol derivative, which was detected before in Norway spruce bark [40]; although the exact compound is not known, the fragmentation pattern found by us (see Table 3) correlates to the suggested derivative (647, 485, 243). Peak 8a is a possible diethyl maleate, coming from lignin cleavage by oxygen and ethanol [41], but no fragmentation pattern could be found in the literature.

Nine compounds were found in the targeted analysis, but in too low a quantity to be detected as distinct peaks by DAD. They are listed in Table 3.

### 2.2. Polyphenolic Compounds in Debarking Water, Bark Press Water, and Bark Extracts

Samples of debarking water and bark press water obtained from the pulp and paper industry were measured by LC-MS/MS. The identities of 12 compounds (6 in a high enough concentration to be detected by the DAD and 6 under the DAD quantification limit) were confirmed in the targeted analysis by the detected mass and fragmentation patterns: quinic acid, protocatechuic acid isomers, syringaldehyde, catechin, ferulic acid, vanillin, oxyresveratrol, p-coumaric acid, isorhamnetin-O-hexoside, spiraeoside, and eugenol (see Table 4). The data in the table show that the debarking water contained lower concentrations of polyphenolic compounds than the bark press water. This observation is in accordance with the literature [4], which found total polyphenol concentrations of 240 mg/L (calculated in total organic content (TOC)) in debarking water and 2470 mg/L TOC in bark press water. This is a consequence of the water having shorter contact with the bark, while bark press water is the water that is adsorbed into the bark structure and squeezed out with the extracts during the pressing of the bark to dryness. These results show that the debarking water is a less interesting source from which to retrieve polyphenolic compounds than bark press water for simple polyphenolic compounds. Simple polyphenolic compounds do not accumulate in excessive concentrations in the recycled debarking waters, and thus, clarifiers and wastewater treatment plants are effective for their removal. However, bark press water seems to be an interesting source with a much higher variety of polyphenolic compounds in high concentrations for the recycling of polyphenolic compounds.

In the second set of experiments, the extraction efficiencies of different solvent preparations were tested (cold water, hot water, ethanol and ethanol/water (1:1); all common extractants used in pharmaceutical preparations) on bark samples with different extraction times in an ultrasound bath. The polyphenolic content was determined with LC-MS/MS. Similar compounds were found in the bark extracts as in the debarking water and bark press water: quinic acid, protocatechuic acid, syringaldehyde, catechin, ferulic acid, oxyresveratrol, isorhamneti-O-hexoside, and spiraeoside. Extra identified compounds were dihydroxyphenyl acetic acid, aromadendrin rhamnoside, dihydrorobinetin, quercitrin, isorhamneti-O-rhamnoside, and robinetin (Table 5). The first trend that can be observed is that a long extraction time was needed for a slight increase in the extracted polyphenolic compounds in the cold and hot water extractions or ethanol/water extractions. In the ethanol extractions, a distinct increase in the polyphenol concentrations was observed according to the extraction time. The use of ethanol/water extracts resulted in the highest concentrations of polyphenolic compounds.

The use of hot water performs similarly to cold water as an extractant. Using water instead of an ethanol-containing solution resulted in a lower variety of polyphenolic compounds extracted: catechin, oxyresveratrol, isorhamnetin hexoside, spiraeoside, and robinetin were not present in the water extracts. This is generally in accordance with the results in Table 5, where such compounds were not detected or detected in quantities lower than the limit of quantification in debarking water and bark press water. Catechin, however, a compound that was only detected in ethanolic extracts, was also found in bark press water from the pulp and paper industry. 

The results of our research were compared with the data found in the literature. The results of previous studies show that the predominant compounds found in bark from Norway spruce are o-hydroxybenzoic acid, ferulic acid, syringaldehyde, vanillic acid, p-hydroxybenzoic acid, protocatechoic acid, p-coumaric acid, gallic acid, cinnamic acid, quercetin, and catechin [42,43,44]. Several stilbenes were also found in Norway spruce bark: piceaside derivatives, taxifolin and luteolin glucoside, rhamnetin, astringin, isorhapontigenin, and piceatannol [40]. Although most of the main compounds were found in our investigated bark samples, quite a few compounds were not detected or were below the limit of quantification. This can be expected, since hydrophilic and phenolic extractive compounds are rapidly lost after debarking. About 60% of the condensed tannins (CT) and about 26% of the quantified lipophilic compounds can be lost after 2 weeks of storage [45]. This means that for the utilisation of valuable polyphenolic compounds, it is necessary to have a fast supply of material for further processing after debarking. This requires efficiency at all levels of the supply chain to ensure that tree delivery times are kept short. Above that, although the bark and debarking water/bark press water come from two different origins, it can be seen that the same main simple polyphenolic compounds (quinic acid, protocatechuic acid, and ferulic acid) are present in the bark water extracts and the debarking water/bark press water. This means that in all sources, these phenolic acids are present in larger quantities and the least prone to degradation.

### 2.3. DPPH Tests

To determine the radical scavenging activity, the EC50 was calculated, which is defined as the concentration of substrate that causes a 50% loss of DPPH activity. These calculations were made in order to evaluate the antioxidant activity of each group of phenolic compounds in relation to their determined concentrations. Higher values of EC50 mean worse radical scavenging activity. The EC50 was determined for the total sample (0.2 µm filtered samples) [46]. To avoid the influence of larger tannins, which are rather toxic, on the antioxidant potential, the samples were also filtered through a 3000 Da filter. The results are shown in Table 6.

For the EC50 of the total samples, it can be seen that the water/ethanol extracts had the highest radical scavenging activity. This was expected because of its high content of extracted polyphenolic compounds. This was followed by hot water bark extracts and bark press water. The debarking water from the pulp and paper industry using a wastewater treatment plant and the bark cold water extracts with short extraction times or ethanol with longer extraction times had a similar performance. The debarking water from the pulp and paper industry had a really low radical scavenging activity, which was also expected from the low concentration of extracted polyphenolic compounds. An interesting phenomenon is that the radical scavenging ability decreased for the water extracts from bark according to the extraction time, while for ethanol, it increased. The ethanol/water mixture first decreased, then increased again. An increase in the radical scavenging ability was expected with an increase in the extracted polyphenolic compounds from the bark or larger polyphenolic compounds, which break down. A decrease in the radical scavenging activity can happen due to the oxidation or reaction of the polyphenolic compounds with themselves or other compounds.

After filtering the samples through a 3000 Da filter, a completely different situation occurred. For all samples, the radical scavenging performance stayed similar or decreased. Ethanol, which before showed quite a poor performance, had the highest radical scavenging activity. This means that the ethanol extracts contained much higher concentrations of low molecular weight polyphenolic compounds than the water extracts. As with the total samples, the radical scavenging ability decreased for the water extracts from bark according to the extraction time, while for the ethanol, it increased. The ethanol/water mixture first decreased, then increased again. Bark press water still had a good performance and scored better than the cold and hot water bark extracts and even the ethanol/water extracts. The debarking water barely contained any radical scavenging activity. The big difference between the filtered and the total sample of debarking water from the pulp and paper mill with a wastewater treatment plant means that mainly larger phenolic compounds were present (lignin and tannins).

## 3. Materials and Methods

### 3.1. Materials and Instrumentation

Chemicals for extraction: deionised water (18.2 MΩ) and ethanol (Carlo Erba, absolute anhydrous for analysis—reagent grade). Solvents for LC-MS/MS analysis: acetonitrile, MeOH and water (Honeywell, LC-MS CHROMASOLV grade). Oxyresveratrol (≥97%, Sigma Aldrich, St Louis, MO, USA), syringaldehyde (98%, Sigma Aldrich, St Louis, MO, USA), spiraeoside (≥95%, Supelco, Merck, Darmstadt, Germany), p-coumaric acid (≥98%, Sigma Aldrich, St Louis, MO, USA), and vanillin (99%, Sigma Aldrich, St Louis, MO, USA) were used as analytical standards to confirm the identities of the peaks and quantification. 2,2-Diphenyl-1-picrylhydrazyl (Merck, Darmstadt, Germany) was used for the determination of the radical scavenging activity, with gallic acid (97.5–102.5%, Sigma Aldrich, St Louis, MO, USA) as an internal standard. The extracts were filtered with 200 nm polyamide (nylon) syringe filters before the LC-MS/MS measurements.

The high-performance liquid chromatography coupled to electrospray ionisation and quadrupole time-of-flight mass spectrometer (HPLC-ESI-QTOF-MS, 6530 Agilent Technologies, Santa Clara, CA, USA) was used to qualify and quantify the present polyphenolic compounds. The HPLC equipment incorporated a Poroshell 120 column (EC-C18; 2.7 µm; 3.0 × 150 mm). The radical scavenging activity was measured using a DPPH assay and determined at 515 nm using an Infinite F200 microplate reader (Tecan, Männedorf, Switzerland).

### 3.2. Sample Collection

Debarking water and bark press water were sampled from a Nordic pulp and paper mill (the source wants to stay unknown). In the pulp and paper mill, the water was used in a Norwegian spruce debarking plant, which was recirculated from a primary clarifier (removal of solids from wastewater from that debarking plant plus the wastewater from a debarking plant that uses pine and birch trees). In the second spruce debarking plant of the Norwegian pulp and paper mill, water was used that was recycled after being sent to a wastewater treatment plant. Sampling was performed in February 2021. The samples were sent frozen and immediately stored in a freezer at −20 °C. Dry bark waste from Norwegian spruce was obtained from the Slovenian sawmill company Gozdno gospodarstvo Postojna d.d. Right before the extraction and analysis, the bark was milled into particles smaller than 1 mm.

### 3.3. Sample Collection

Milled bark (0.5 g) was added to 20 mL of solvent (cold water (CW), ethanol (E), cold water/ethanol (1:1) (CWE), and hot water (HW) (see Table 7). The samples were placed in an ultrasound bath for 2, 5, or 30 min. The samples were filtered through 0.2 µm filters before LC-MS-MS analysis.

### 3.4. LC-MS/MS Analysis

HPLC-ESI-Q-TOF-MS: An elution gradient of 100% water/formic acid (99.5:0.5, *v/v*) (A) towards 100% acetonitrile/MeOH (1:1, *v/v*) was used over a period of 20 min (flow rate: 0.5 mL/min; injection volume: 1 µL). The separated phenolic compounds were firstly monitored using a diode-array detector (DAD) (280 nm), and then MS scans were performed in the *m/z* range of 40–1000 (capillary voltage: 2.5 kV; gas temperature: 250 °C; drying gas: 8 L/min; sheath gas temperature: 375 °C; sheath gas flow: 11 L/min). In these conditions, the instruments were expected to provide experimental data with accuracy within ±3 ppm. All data were processed using Qualitative Workflow B.08.00 and Qualitative Navigator B.080.00 software.

The extracts were screened for the range of phenolic compounds previously reported in the literature for bark and wood, and their identifications were confirmed, based on accurate mass and fragmentation profiles, with the literature data and analytical grade standards (oxyresveratrol, syringaldehyde, and spiraeoside).

The quantification of the total phenol concentrations in the samples was performed using calibration graphs prepared using oxyresveratrol by HPLC-DAD. The calibration plots indicate good correlations between the peak areas and the commercial standard concentrations. The regression coefficients were higher than 0.990. The limit of quantification (LOQ) was determined as the signal-to-noise ratio of 10:1 and was 8.3 µg/mL. For individual polyphenolic compounds found by MS, only semi-quantification was possible, since the standards of all compounds are needed for full quantification.

### 3.5. Radical Scavenging Activity Measured Using DPPH Assay

Bark extracts, debarking water, and bark press water were filtered using 0.2 µm filters. A part of this sample was then again filtered through 3000 Da filters to remove the larger phenols, which have no influence on human health, so the anti-oxidant activity can be compared.

The antioxidant activity of the different extracts was assessed using the radical scavenging ability in the 1,1-diphenyl-2-picrylhydrazyl (DPPH) radical assay and conducted as reported by Žegura et al. [47] with modifications, including the replacement of methanol with ethanol and the use of gallic acid, rather than ascorbic acid, as a standard for a positive control. Reaction mixtures containing 100 μL of differently diluted extracts and 100 μL of 0.2 mM DPPH in methanol were incubated for 60 min in darkness at ambient temperature, using 96-well microtiter plates. The decrease in the absorbance of the free radical DPPH was measured at 515 nm with a microplate reader. The free radical scavenging activity was calculated as the percentage of DPPH radical that was scavenged, and the details are explained elsewhere [48]. The EC50 value concentrations at 50% of DPPH radical scavenged were determined graphically from the curves. Two independent experiments with two replicates each were performed. The DPPH was chosen as a quick method for measuring the oxidation of the polyphenolic compounds in the debarking water, bark press water, and bark extracts. These measurements enabled us to correlate the polyphenolic quantities, obtained by LC-MS-MS, with their antioxidant activity. Consequently, a more calculated decision could be made about which samples would be of interest for further in vivo studies.

## 4. Conclusions

The polyphenolic compositions of bark extracts, debarking water, and bark press water were determined by LC-MS-MS analysis. When bark was extracted with water instead of a solvent containing ethanol, fewer polyphenolic compounds were released, but the amounts were still high enough to be an appropriate source of polyphenols for industrial applications. Although the bark press water did not have such a versatile range of polyphenolic compounds as the bark extracted with water, its composition was fairly close to the latter, and thus it could be a potential source of smaller polyphenolic compounds for the specialty chemical sector. In both the bark water extracts and bark press water, similar compounds were identified, as found by other studies on spruce bark. However, certain polyphenolic compounds were not identified. This can be expected since industrial bark waste was used, and polyphenolic concentrations rapidly decrease within days after logging trees. This observation points to the need to process bark in as short a time as possible if it needs to be used as a rich-as-possible source of polyphenolic compounds for other industries. The antioxidant activity and high polyphenol concentrations show that bark press water is a good source of polyphenolic compounds. Debarking water is less suitable since it has lower polyphenolic concentrations and especially a low antioxidant activity. Further research is needed to find economically suitable ways to collect the polyphenolic compounds from bark press water. Until now, mainly adsorption techniques are used with the use of adsorption beds, but other techniques, such as the modification of magnetic iron particles, may be an interesting alternative since they could be easily implemented in existing infrastructure.

## Figures and Tables

**Figure 1 molecules-28-00542-f001:**
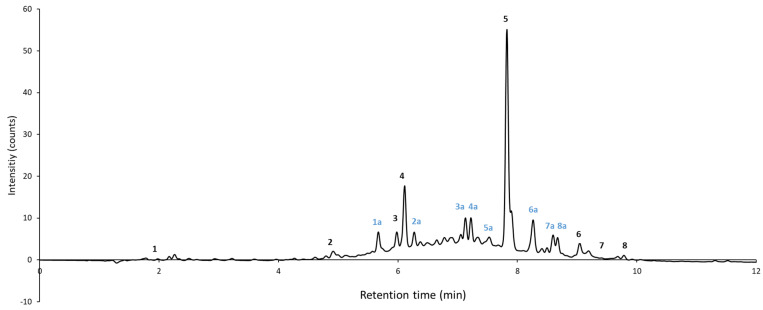
Elution profile of polyphenolic compounds in bark extract. Peaks determined with targeted analysis are numbered in black (1, 2, 3, etc.). Peaks on which untargeted analysis was tested are numbered in blue (1a, 2a, 3a, etc.).

**Table 1 molecules-28-00542-t001:** Identity of discernible peaks found with DAD analysis combined with targeted mass spectrometry.

Peak No.	Retention Time	Identity Compound	*m/z*	Detected Fragments	Fragments Found in Literature
1	2.0	Quinic acid	191.0555	173, 127, 111, 93, 87, 85	173, 127, 111, 85
2	4.9	Protocatechuic acid	153.0182	153, 109	153, 109
3	6.0	Catechin	289.0713	289, 245, 205, 203, 179, 125, 109	289, 271, 245
4	6.1	Ferulic acid	193.05	193, 178, 149, 134	178, 149, 134
5	7.8	Dihydrorobinetin	303.0506	303, 285	303, 285
6	9.0	Quercitrin	447.0927	301	300, 301, 271, 255, 151
7	9.4	Spiraeoside	463.0873	301	301
8	9.8	Isorhamnetin rhamnoside	461.1086	315	315

**Table 2 molecules-28-00542-t002:** Discernible peaks in DAD analysis that were not identified with targeted analysis. A possible identity was given with untargeted analysis.

Peak No.	Retention Time	*m/z*	Detected Fragments	Possible Identity Compound
1a	5.7	371.0970	325, 205, 163, 119, 45	Oxomatairesinol
2a	6.3	443.1320	443, 113, 101, 89, 71, 59	Unknown
3a	7.3	451.1248	406, 405, 243, 201, 199, 173, 159	(Epi)catechin-*O*-glucoside
4a	7.3	441.0954	441, 405, 243, 201, 159, 113, 112,	(Epi)catechin monogallate
5a	7.5	525.1978	327, 179, 167, 161, 146, 134, 119, 89	Isorhamnetin glucoside
6a	8.3	373.1506	373, 281, 161, 143, 101, 45	(Iso)hdydroxymatairesinol
7a	8.5	647.1764	647, 605, 600, 485,435, 401, 361, 309, 241	Piceatannol derivative
8a	8.8	171.0734	171, 145, 127, 103, 83	Diethyl maleate

**Table 3 molecules-28-00542-t003:** Polyphenolic compounds identified with targeted analysis by MS but not detected by DAD.

Retention Time	Identity Compound	*m/z*	Detected Fragments	Fragments in Literature
4.8	Syringaldehyde	181.0499	181, 151	181, 166, 151
5.2	Dihydroxyphenylacetic acid	167.0342	123	123
6.7	Vanillin	151.0391	151, 136, 108, 107	151, 137, 136, 123, 108, 107
6.2 and 7.2	Oxyresveratrol	243.0652	225, 199, 157	225, 199, 157, 133, 115
7.8	P-coumaric acid	163.0395	163, 119	163, 119, 94
7.8 and 8.1	Aromadendrin-rhamnoside	433.1148	269, 179, 151	287, 269, 259, 180, 179, 151
9.2	Isorhamnetin-O-hexoside	477.1043	315	315
10.4	Robinetin	301.0359	301, 273, 245	301, 273, 245, 229, 135, 91
13.2	Eugenol	209.0808	163	163, 149, 147, 137

**Table 4 molecules-28-00542-t004:** Phenolic compounds found in debarking water (D) or bark press water (P) obtained from the pulp and paper industry using a clarifier (C) or wastewater treatment plant recycling (W). The quantities for each compound are given as peak areas. The total polyphenol concentration is given in mg/mL.

Compound (RT in min)	PC	PW	DC	DW
Quinic acid	85,989 ± 1912	81,331 ± 2084		83,169 ± 2719
Protocatechuic acid (isomer 1)	<LOQ	<LOQ		
Syringaldehyde	6539 ± 581	<LOQ	<LOQ	<LOQ
Protocatechuic acid (isomer 2)	16,204 ± 1184	23,408 ± 695	<LOQ	11,451 ± 1292
Catechin	8461 ± 1419	>LOQ		
Ferulic acid	13,006 ± 523	15,998 ± 767		9151 ± 1304
Vanillin	<LOQ			
Oxyresveratrol		<LOQ		
*P*-coumaric acid	10,106 ± 2511			<LOQ
Isorhamnetin-O-hexoside		<LOQ		
Spiraeoside		<LOQ		
Eugenol		6431 ± 727		9070 ± 247
Total polyphenol concentration (mg/mL)	12.99 ± 0.22	15.55 ± 0.81	1.95 ± 0.02	4.76 ± 0.35

**Table 5 molecules-28-00542-t005:** Phenolic compounds found in bark extracts, which were obtained by extraction for 2, 5, or 30 min in an ultrasound bath with different solvents (cold water, ethanol, ethanol/water, or hot water). The quantities for each compound are given as peak areas. The total polyphenol concentration is given in mg/mL.

Compound	Bark Cold Water Extraction			Bark Ethanol/Water (1:1) Extraction			Bark Ethanol Extraction			Bark Hot Water Extraction		
(RT in min)	2 min	5 min	30 min	2 min	5 min	30 min	2 min	5 min	30 min	2 min	5 min	30 min
Quinic acid	91,055 ± 5082	88,490 ± 4899	97,538 ± 5933	73,973 ± 4246	80,280 ± 1328	81,208 ± 2909	3734 ± 817	6093 ± 451	8290 ± 633	87,926 ± 2482	84,382 ± 4714	90,067 ± 570
Syringaldehyde	<LOQ	<LOQ			<LOQ					<LOQ		<LOQ
Protocatechuic acid	8274 ± 778	9894 ± 1117	11,650 ± 581	9432 ± 886	10,745 ± 214	10,132 ± 93	1011 ± 14	2090 ± 381	3696 ± 871	10,773 ± 361	11,477 ± 424	14,670 ± 1067
Dihydroxyphenylacetic acid		<LOQ										
Catechin				17,701 ± 729	17,458 ± 1583	18,348 ± 1169	10,004 ± 670	13,353 ± 567	16,185 ± 105			<LOQ
Ferulic acid	7771 ± 857	7388 ± 416	9443 ± 220	7557 ± 261	6633 ± 871	7490 ± 85		<LOQ	<LOQ	7731 ± 80	8325 ± 763	8043 ± 57
Aromadendrin-rhamnoside				<LOQ								
Dihydrorobinetin	8214 ± 65	9153 ± 2554	16,081 ± 522	42,783 ± 4072	47,511 ± 3313	49,642 ± 894	20,044 ± 1785	27,636 ± 3964	44,938 ± 1006	10,325 ± 3840	14,068 ± 5642	14,914 ± 2224
Oxyresveratrol			<LOQ	15,509 ± 2885	17,221 ± 3324							
Quercitrin	3582 ± 911	3902 ± 784	5816 ± 419	15,565 ± 1036	16,300 ± 732	17,460 ± 296	5003 ± 834	6667 ± 323	9063 ± 596	4549 ± 299	5775 ± 192	7951 ± 94
Isorhamnetin-hexoside	<LOQ			<LOQ							<LOQ	
Spiraeoside			<LOQ	29,498 ± 1639	30,412 ± 4680	30,695 ± 1565	14,107 ± 2418	17,537 ± 695	22,268 ± 671			<LOQ
Isorhamnetin-O-rhamnoside	<LOQ		< LOQ	9333 ± 179		9493 ± 503			<LOQ	3243 ± 24	3873 ± 53	5688 ± 88
Robinetin				1089 ± 217	1404 ± 139	1528 ± 136						
Total polyphenol concentration (mg/mL)	7.24 ± 1.11	7.07 ± 1.75	9.21 ± 0.55	15.86 ± 2.27	16.35 ± 1.62	16.74 ± 0.30	6.84 ± 0.41	9.39 ± 0.47	13.18 ± 2.03	7.54 ± 0.05	7.29 ± 0.01	9.41 ± 0.64

**Table 6 molecules-28-00542-t006:** DPPH test of bark (CW: cold water extraction; E: ethanol extraction; HW: hot water extraction with 2, 5, or 30 min extraction time), and debarking water (D) and bark press water (P) from the pulp and paper industry using a water recycling system with clarifier (C) or wastewater treatment plant (W). Samples were filtered through standard 0.2 µm filters or 3000 Da filters. The results are expressed as the gallic acid equivalent antioxidant capacity (GAE) and as equivalent concentrations of test samples scavenging 50% of DPPH radical (EC50).

Sample	0.2 µm Filtered Samples		3000 Da Filtered Samples	
	EC50 (µg/mL)	mg GAE/g (Dry Mass)	EC50 (µg/mL)	mg GAE/g (Dry Mass)
CW2	154 ± 25	3.23 ± 0.97	560 ± 28	1.26 ± 0.06
CW5	203 ± 16	2.24 ± 0.16	920 ± 46	0.48 ± 0.02
CW30	213 ± 13	1.96 ± 0.22	971 ± 49	0.42 ± 0.02
CWE2	77 ± 24	6.90 ± 2.06	425 ± 21	1.02 ± 0.05
CWE5	105 ± 4	4.27 ± 1.70	450 ± 23	0.49 ± 0.02
CWE30	85 ± 16	4.54 ± 1.27	432 ± 22	0.51 ± 0.03
E2	317 ± 21	1.26 ± 0.05	332 ± 17	1.38 ± 0.07
E5	238 ± 81	1.31 ± 0.81	166 ± 8	2.26 ± 0.11
E30	161 ± 13	1.84 ± 0.99	153 ± 8	1.41 ± 0.07
HW2	108 ± 5	5.27 ± 0.90	584 ± 29	0.86 ± 0.04
HW5	113 ± 16	4.13 ± 0.23	901 ± 45	0.51 ± 0.03
HW30	167 ± 66	2.40 ± 0.62	1030 ± 52	0.34 ± 0.02
PC	134 ± 1	4.02 ± 0.70	391 ± 20	1.73 ± 0.09
PW	136 ± 5	3.53 ± 0.15	391 ± 20	1.04 ± 0.05
DC	3333 ± 590	0.17 ± 0.04	3006 ± 150	0.18 ± 0.01
DW	169 ± 18	1.62 ± 0.12	1340 ± 67	0.24 ± 0.01

**Table 7 molecules-28-00542-t007:** Conditions for the extraction of polyphenolic compounds from bark with cold water (CW), ethanol (E), and hot water (HW) at different extraction times.

Sample	Temperature	Time (min)	Solvent
CW2	T_R_	2	Distilled water
CW5	T_R_	5	Distilled water
CW30	T_R_	30	Distilled water
CWE2	T_R_	2	Deionised water and ethanol (50:50)
CWE5	T_R_	5	Deionised water and ethanol (50:50)
CWE30	T_R_	30	Deionised water and ethanol (50:50)
E2	T_R_	2	Ethanol
E5	T_R_	5	Ethanol
E30	T_R_	30	Ethanol
HW2	80 °C	2	Deionised water
HW5	80 °C	5	Deionised water
HW30	80 °C	30	Deionised water

## Data Availability

Not applicable.

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
