# Peer review of "The Potential Value of Debarking Water as a Source of Polyphenolic Compounds for the Specialty Chemicals Sector"

_molecules, 2023, doi:10.3390/molecules28020542_

Round 1

Reviewer 1 Report

Submited manuscript „The Potential Value of Debarking Water as a Source of 2 Polyphenolic Compounds for the Specialty Chemicals Sector“ authores: Kelly Peeters , Esakkiammal Sudha Esakkimuthu , Črtomir Tavzes ,Katja Kramberger and Ana Miklavčič  Višnjevec,  is a clearly written and well-conceived paper. Isolation and identification of chemical compounds is well done. I have no objections to that part. In conclusion the authors state that „Further research is needed to find economical suitable ways to collect the polyphenolic compounds from bark press water“. I would ask the authors to comment on this in a little more detail.

I suggest that the paper be accepted as submitted in a small comment for potential financially profitable  application.

Author Response

The authors want to thank reviewer 1 for the positive feedback. We adapted the conclusions according to the feedback of both reviewers in a compact paragraph, with some elaboration on future research.:

The polyphenolic composition of bark extracts, debarking water and bark press water was determined by LC-MS-MS analysis. When bark was extracted with water instead of a solvent containing ethanol, less polyphenolic compounds are released, but the amounts seem still high enough to be appropriate source of polyphenols for industrial applications. Although bark press water has not such a versatile range of polyphenolic compounds as bark extracted with water, its composition is fairly close to the latter, and thus can be a potential source of smaller polyphenolic compounds for the specialty chem-ical sector. In both bark water extracts and bark press water, similar compounds were identified as found by other research in spruce bark. Certain polyphenolic compounds were however not identified. This can be expected since industrial bark waste was used, and polyphenolic concentrations rapidly decrease within days after logging trees. This observation points to the need to process bark in an as short as possible time if it needs to be used as an as rich as possible source of polyphenolic compounds for other industries. The antioxidant activity and high polyphenol concentrations showed that bark press water is a good source for polyphenolic compounds. Debarking water is less suitable since it has lower polyphenolic concentrations and especially a low antioxidant activity. Further research is needed to find economical suitable ways to collect the polyphenolic com-pounds from bark press water. Until now, mainly adsorption techniques are used by the use of adsorption beds but other techniques such as the modification of magnetic iron particles can become an interesting alternative, since it could be easily implemented in existing infrastructure.

Reviewer 2 Report

This is an interesting article addressing the forest-based industries waste usage possibilities, by considering its economic potential due to the present polyphenolic compounds, useful bioactive compounds of commercial interest of various sectors. This article is focusing on the extraction methods, identification of the molecular structure and determination of the antioxidant capacity of polyphenols. The results showed that considerable amount of low molecular polyphenolic compounds of interest for high-end applications compounds were found in all obtained extracts regardless the solvent. The antioxidant activity and high polyphenol concentrations highlighted the bark press water as a good source for polyphenolic compounds, and the authors pointed out that further research is needed to find economical suitable ways to collect those polyphenols.

I’ve found the article research background adequately described. The used methods and the research results are well presented and discussed. The reference list is relevant, although not entirely current. English language and style are generally satisfying. There are only a few minor observations to be addressed.

Table 1, 2 – Peak number or Peak No., instead of Peak nr

Table 4, 5 – Check the expression >LOQ, should it be <LOQ? If it is >LOQ, then there should be some numerical values as the others in the table.

Conclusion – I advise you to write the conclusion as a compact paragraph without graphic marks.

Author Response

The authors want to thank reviewer 2 for the positive feedback. The tables were corrected according to reviewers’ 2 suggestions. The authors want to thank reviewer 2 for noticing the errors. We adapted the conclusions according to the feedback of both reviewers in a compact paragraph, with some elaboration on future research.:

The polyphenolic composition of bark extracts, debarking water and bark press water was determined by LC-MS-MS analysis. When bark was extracted with water instead of a solvent containing ethanol, less polyphenolic compounds are released, but the amounts seem still high enough to be appropriate source of polyphenols for industrial applications. Although bark press water has not such a versatile range of polyphenolic compounds as bark extracted with water, its composition is fairly close to the latter, and thus can be a potential source of smaller polyphenolic compounds for the specialty chem-ical sector. In both bark water extracts and bark press water, similar compounds were identified as found by other research in spruce bark. Certain polyphenolic compounds were however not identified. This can be expected since industrial bark waste was used, and polyphenolic concentrations rapidly decrease within days after logging trees. This observation points to the need to process bark in an as short as possible time if it needs to be used as an as rich as possible source of polyphenolic compounds for other industries. The antioxidant activity and high polyphenol concentrations showed that bark press water is a good source for polyphenolic compounds. Debarking water is less suitable since it has lower polyphenolic concentrations and especially a low antioxidant activity. Further research is needed to find economical suitable ways to collect the polyphenolic com-pounds from bark press water. Until now, mainly adsorption techniques are used by the use of adsorption beds but other techniques such as the modification of magnetic iron particles can become an interesting alternative, since it could be easily implemented in existing infrastructure.